# Wireless Charging Deployment in Sensor Networks

**DOI:** 10.3390/s19010201

**Published:** 2019-01-08

**Authors:** Wei-Yu Lai, Tien-Ruey Hsiang

**Affiliations:** Department of Computer Science and Information Engineering, National Taiwan University of Science Technology, Taipei 10607, Taiwan; d10115005@mail.ntust.edu.tw

**Keywords:** wireless rechargeable sensor network, charger planning, number of charging stops, charging time

## Abstract

Charging schemes utilizing mobile wireless chargers can be applied to prolong the lifespan of a wireless sensor network. In considering charging schemes with mobile chargers, most current studies focus on charging each sensor from a single position, then optimizing the moving paths of the chargers. However, in reality, a wireless charger may charge the same sensor from several positions in its path. In this paper we consider this fact and seek to minimize both the number of charging locations and the total required charging time. Two charging plans are developed. The first plan considers the charging time required by each sensor and greedily selects the charging service positions. The second one is a two-phase plan, where the number of charging positions is first minimized, then minimum charging times are assigned to every position according to the charging requirements of the nearby sensors. This paper also corrects a problem neglected by some studies in minimizing the number of charging service positions and further provides a corresponding solution. Empirical studies show that compared with other minimal clique partition (MCP)-based methods, the proposed charging plan may save up to 60% in terms of both the number of charging positions and the total required charging time.

## 1. Introduction

Different core wireless charging technologies have been developed, which can be classified into inductive coupling, electromagnetic (EM) radiation, and magnetic resonant coupling [1]. Inductive coupling provides highly efficient charging for a single target at very close range. It is often applied when charging mobile phones, electric toothbrushes, and other devices. The charging efficiency of EM radiation decreases with distance, but it can be used in wireless sensing networks owing to the small size of the receiving power device. Magnetic resonant coupling provides high-efficiency charging for targets within a few meters and is less susceptible to weather factors, making it suitable for wireless sensing networks, mobile devices, and electric vehicles. For wireless charging technology, a wireless sensing network is a common application environment.

With the advances in wireless charging technology in recent years, the combination of wireless sensor networks and mobile wireless chargers can prolong the lifespan of the wireless sensor network. A sensor node is responsible for the communication within the network, and its lifespan is determined by its batter power. Thus, when the sensor battery is exhausted, the network may be interrupted. In past studies, network communication was maintained mostly by saving the energy consumption of the sensors. As the range and efficiency of wireless charging have improved, there have already been studies discussing the use of wireless charging technology for sensor networks to charge the sensors with a mobile wireless charger, thereby avoiding battery drain of the sensors. While mobile wireless charger could help extend the life of wireless sensor networks, planning the charging path for these devices has become a priority.

When planning the moving path of a mobile charger, the essential locations, called the *charging stops*, that the device must stay and provide service must be selected. As a charger can charge multiple sensors simultaneously, it is not necessary to pass all the sensors. However, the charging efficiency is limited by the distance between the charger and the sensor, and hence, the charging stops are set first and subsequently, the moving route through the stops is planned.

Besides selecting the positions of charging stops, a charger must determine the time period, called the *charging time*, that it must stay at each stop to provide service. Due to the charger does not necessarily provide sufficient power to the sensor all at once, it needs to wait at each charging stop for a period of time to provide sufficient power for the sensor. Furthermore, the *total charging time* of a charger is defined by the sum of charging times over its charging stops.

When determining the charging time at a stop, a charger usually must consider the charging efficiency and the amount the power requested by sensors. In most research the charging efficiency is considered not affected by the number of sensors [2,3,4,5,6,7,8,9,10]. In other words, the time required to charge multiple power-depleted sensors is the same as to charge a single one. As a consequence, the number of charging stops and the total charging time are often considered the main cost of the charging path.

Although the number of charging stops and the total charging time are regarded as the main cost of the charging path, most of the current studies only considered either the minimum number of charging stops or the total charging time. They did not consider that after minimising the number of charging stops, it was possible to minimise the total charging time further, and the reverse is also true. If the objective is only to minimise the number of charging stops, there may be many different setups. Although different setups may have the same number of stops, the total charging time may not necessarily be the same. For example, in Figure 1, there are four sensors that need to be charged. Among them, *s*_1_ and *s*_4_ need one second, whereas *s*_2_ and *s*_3_ both require three seconds. The sensor’s location distribution allows a charger at a stop to simultaneously charge at most two sensors. Moreover, *s*_2_ and *s*_4_ cannot be charged simultaneously. It can be observed that in this case a minimum number of two charging stops must be set up. If the stop is set such that *s*_1_ and *s*_2_ are charged simultaneously, and *s*_3_ and *s*_4_ are charged simultaneously, the charging time of both charging stops will be three seconds. On the other hand, if *s*_1_ and *s*_4_ are simultaneously charged and *s*_2_ and *s*_3_ are simultaneously charged, the total required charging time will be four seconds. Therefore, when planning the stops, besides considering the number of stops, it is necessary to consider the distribution of the stops to further reduce the energy and time required for the charging path. 

In addition to considering setting the locations of the charging stops, as the charging time required for each sensor may vary, the total charging time can be reduced if the charging time at each charging stop is properly planned. When a mobile charger moves to a charging stop, if it only moves to the next charging stop after all the sensors are fully charged in its charging range, it may take more time to charge the sensors. Considering Figure 2 as an example, for charging stops *l*_1_ and *l*_2_ in the figure, sensor *s*_1_ needs six seconds to complete charging, whereas sensors *s*_2_ and *s*_3_ only need three seconds. When the charger arrives at *l*_1_ first, if it fully charges *s*_1_ and *s*_2_ and subsequently moves to *l*_2_ to charge *s*_3_, nine seconds will be required. However, the best way to allocate the charging time is to stop at *l*_1_ and *l*_2_ for three seconds each.

In summary, in addition to considering the minimum number of charging stops, this study considered minimising the total charging time while setting up the minimum number of stops. The main contributions of this paper are as follows: 1)This paper proposed an integer linear programming model, whose goal is to set the minimum number of stops and subsequently minimise the total charging time. In the common integer programming model, the optimisation objective is usually to minimise either the number of stops or the total charging time. However, the proposed model considered both the number of charging stops and the total charging time. In addition, the proposed charging path planning problem was divided into two sub-problems of minimum number of charging stops and minimum charging time, which are discussed in this paper.2)In the problem of minimum number of charging stops, the applicability of the minimal clique partition was analysed. It was observed that a clique required at least three stops be set up to complete the charging. Moreover, the more sensors there were in the clique, the higher the probability of requiring the setup of several stops. Through analysis, it was observed that the stop setup method based on minimal clique partition in the past studies was more suitable for the low-density and short-distance charging environment.3)For the charging planning problem proposed in this paper, a 2-phase planning strategy was proposed. The maximum differences between the planning result of this strategy and the best planning were analysed. In the first phase, the objective was to minimise the number of charging stops, and the number of planned stops should be no more than five times the optimal number. In the second phase, the charging time at each charging stop was set with the objective of minimising the total time. Based on the planning results of the first phase, the total charging time set in the second phase should be no more than three times the minimum total time.4)In addition to the 2-phase setup strategy, a planning strategy was proposed based on the lower bound of the optimal solution. The maximum differences between the planned outcomes and the minimum number of stops as well as the minimum total charging time were analysed. A sensor was selected in this planning strategy. The number of charging stops and charging time were planned according to the location and the required charging time of the sensor. It was guaranteed that the total number of charging stops and the total charging time would not exceed seven times the best planning results.5)As it is abnormal to generate the worst planning results, the simulation results were provided and compared with the results of related studies. In the simulation experiment of the minimum number of charging stops problem, the methods mainly included the 2-phase setup strategy, minimal clique partition-based planning method, greedy algorithm, and integer linear programming. For the minimum number of charging stops problem, it can be observed through analysis and simulation that the proposed method was more suitable for the high-density and long-distance charging environment compared with the methods in previous studies. In addition, the total charging time of the proposed method was lower than that in previous studies.

## 2. Related Work

Wireless sensor network applications have received wide attention in the past years [11,12]. The lifespan of a wireless sensor network is crucial to the success of an application. Therefore, effective charging strategies are needed for the sensors. Furthermore, when mobile chargers are available, it is necessary to optimize their charging paths to reduce the operation costs.

When planning the charging path, many studies consider the case where a charger can only one-by-one charge each sensor. In addition to optimisation objectives related to the charging path, optimisation objectives related to the sensors were considered in some studies. Zou et al. [13] considered deploying a vehicle with *k* chargers and placing them at the locations of the sensors. Once the sensors were fully charged, the chargers were collected by the vehicle and subsequently provided to another *k* sensors. The charging path was planned with the shortest moving distance of the vehicle as the objective, and the chargers could charge all the sensors. Zou et al. [13] first determined the total path through all the sensors using the TSP polynomial time approximation algorithm. The total path was subsequently divided into sub-paths every *k* sensors. The total path was divided into several sub-paths. Once it reached the end of each sub-path, the charger directly returned to the starting point of the sub-path and subsequently moved to its end point, and thereafter moved to the starting point of the next sub-path. Therefore, the path within the sub-path could be divided into three categories. As the path length from the starting point to the end point was the longest among the above three, the moving distance of the charger would not be longer than three times the total path. Therefore, for the shortest path planning problem where only *k* sensors could be charged per trip, Zoe et al. [13] proposed a planning method with an adjustable approximation coefficient. In addition to considering the moving path length of the charger, they considered the objective of minimising the longest electricity drain time and proposed two heuristic path planning methods. Fu et al. [14] believed that a shorter charging time for sensors indicated a higher charging efficiency. Therefore, in order to improve the charging efficiency and shorten the moving distance of the charger, Fu et al. [14] grouped the sensors according to their power consumption rate. Each time, the charger would charge the sensors in the same clique, thereby reducing the charging time of sensors and the moving distance of the charger. In the surveillance area, each location was not necessarily monitored by only one sensor. Shu et al. [15] used the coverage of the sensors as the network life. If there were locations in the surveillance area not detected by a sensor owing to the sensor’s power being exhausted, the network life was considered to be exhausted. Under the energy limitation of the charger, the charging schedule of the sensor was established through linear programming to maximise the network life.

Although the positions of sensors can be used as candidates of charging stops, there are cases where it is impossible for a mobile charger to arrive at a sensor’s location. Rao et al. [16,17] considered the situation where the sensors are installed inside a building, therefore the charging stop of a sensor must be specified first to starting planning the charging path. In order to avoid the possibility that a sensor’s power is depleted before it can be charged due to the excess charging requests from other sensors, [17] designed corresponding constraints in their ILP models.

In the case where the charger may simultaneously charge multiple sensors, usually the charger is not required to pass by every sensor so that its charging path can be shortened. When using the minimum charging time as the optimization objective while planning the charging stops, in order to reduce the charging time, the past studies would not limit the charging range of the wireless chargers. However, the amount of power received by the sensor per unit time is related to its distance from the charger. The further the distance, the less power is received per unit time. According to the above relationship between the sensor receiving power and the distance, Suo et al. [4] proposed the problem of minimizing the charging time at the charging stops. The goal was to use the least charging time for each sensor to receive at least an electric power of *δ*. Subsequently, the sensor distribution area was discretized, and the discrete point sets were used as the candidate locations for the charging stops. An integer linear programming model was established accordingly and the locations for the setting up of the charging stops were determined. However, Fu et al. [18] observed that a plan that uses the charging efficiency function would generate too many charging stops. Thus, after determining the charging stops, the *k*-means algorithm was employed to reduce the number of charging stops. Han et al. [6,7] also adopted *k*-means to establish sensor clusters. In each cluster, a cluster head, which is responsible to report cluster information to the charger, is chosen according to the intra-distances and the remaining power levels of sensors in the cluster. Then the charger selects a charging stop for each cluster and a shortest path to go through all charging stops.

In addition, in order to reduce the number of charging stops, Khelladi et al. [2,3] used the unit disk model as the charging range of the charging stops. They also proposed the use of the minimum number of charging stops as the optimisation objective. The auxiliary graph *G*(*V*, *E*) was created according to the unit circle diameter 2*r*. The sensor required to be charged was represented by *v_i_* ∈ *V*. If e*_i_*_, *j*_ ∈ *E*, the distance between sensors *v_i_* and *v_j_* is less than 2*r*. Finally, *G* was partitioned using the minimal clique algorithm. As a charging stop may not be able to charge all the sensors in a clique simultaneously, a minimum number of charging points would be set up for each clique after partition to ensure that every sensor could be charged. After planning the charging stops, the charging time at each charging stop was determined according to the power required by sensors in the charging range. It was observed through simulation that the minimal clique partition-based plan had fewer charging stops than the method proposed by Fu et al. [18], and could also supply the required power for all the sensors. After planning the stops, the shortest path that passed through all the stops was planned through the relevant studies of TSP. Compared to the auxiliary graph *G* used by [2], Lin et al. [8] guaranteed *G* be connected by adding an extra vertex v that connects to every vertex of *G*. In doing so, a minimum connected dominating set, which serves as the basis of sensor clusters, can be obtained from *G*. The charging stops, which are later connected by a charging path, are then chosen from the clusters.

Besides the power limits of sensors, some work also consider the case where the charger can only supply limited power when planning the charging stops. Ma et al. [9] determined the charging benefit according to the remaining power of sensors and greedily planned the charging path with optimal charging benefit. In addition, Ref. [9] required that sensors cannot request more power than the charging capacity of the charger.

In addition to sensor networks, wireless chargers are often used by smart phones. Xu et al. [19] considered the case where remaining power of a phone reflects the level of satisfaction of a user, and every smart phone has a specified charging timeslot. They required that a charger can only serve a fixed number of smart phones at the same time and established a charging path with optimal satisfaction.

Moraes and Har [5] selected a part of the sensors’ locations as the moving targets of the charging path in addition to pre-establishing the candidate locations. Moreover, the minimum charging time required by the charging path was set as the optimisation objective. Circles of radius *r* with the sensors’ locations as the centres were established. Each circle was regarded as a candidate clique. The centre of each circle was considered as the leader of the candidate clique. The sensors within the circle were the members of the candidate clique. After the candidate cliques were established, the ones with the most clique members were selected sequentially until all the sensors were in cliques. The moving path of the charger was subsequently planned according to the clique leaders. When planning the moving path, the GSS [20] algorithm was used to determine the moving path of the charger from the current location to the location of the next clique leader. After deciding the charging stops, Xie et al. [21,22] considered the power consumption speed and planned the charging path and the charging time at each charging stop. They planned the charging path in order to obtain the maximum break time of a charger.

There are also studies where the working path of the charger is beyond 2D. Zorbas and Douligeris [10] considered using drones as wireless chargers, where the sensors were first grouped in a greedy fashion, then the center of the minimal enclosing circle was utilized in planning a charging path in 3D.

## 3. Sensor Model and Problem Description

It was assumed that there were a total of *n* sensors *S* = {*s*_1_, *s*_2_, …, *s_n_*} that needed to be charged. The power receiving range of every sensor was *r* and the charging efficiency was not affected by distance. The charging time required by sensors was represented by *T* = {*t*_1_, *t*_2_, …, *t_n_*}. For sensor *s_i_* ∈ *S*, the required charging time is *t_i_* ∈ 
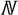
.

Based on the related description of the aforementioned sensors, the candidate locations of charging stops *L* = {*l*_1_, …, *l_m_*} were set up according to the locations of sensors and the charging distances, and *m* ≦ 2*n*^2^. When sensor *S*’ ⊆ *S* was in the circle of radius *r* with *p* as the centre, if the distance between every *s_i_* ∈ *S*’ and *p* was less than *r*, then a *p*’ could be determined such that the distance between *s_j_*, *s_k_* ∈ *S*’, and *p*’ was equal to *r*, and the distance between *s_i_* and *p*’ was less than *r*. Therefore, circles of radius *r* were established with the location of sensor *s* as the centre. The intersections of the two circles were considered as the candidate locations for the charging stops, as shown in Figure 3. 

In addition to considering the locations of the stops, the charging time at each stop was considered. Thus, *l’_i_*_,*j*_ ∈ *L*’ was used to represent waiting *j* units of time at the candidate location *l_i_*. If *l’_i_*_,*j*_ was selected as the charging stop and the charging time, then it was denoted as *l’_i_*_,*j*_ = 1; otherwise, *l’_i_*_,*j*_ = 0. The maximum charging time required by the sensor was *z* = max (*t_k_* ∈ *T*), and hence, the maximum charging time at each stop was *z*, and |*L*’| = *mz*. After placing the candidate stops, *a_ij_* = 1 (*a_ij_* = 0) was used to indicate that the distance between the candidate location *l_i_* and the sensor *s_j_* was less than (more than) *r*. The proposed charging stops planning model is as follows: (1)$$\min\overset{z}{\underset{k=1}{\sum }}\overset{m}{\underset{i=1}{\sum }}l_{i,\;k}+\frac{l\prime_{i,\;k}k}{\left(m+1\right)z}$$
subject to:(2)$$\overset{z}{\underset{k=1}{\sum }}\overset{m}{\underset{i=1}{\sum }}a_{ij}l\prime_{i,\;k}k\geq t_{j}$$
∀*j* ∈ {1, …, *n*}(3)
*l’_i_*_, *k*_ = {0, 1}(4)
*t_j_* ∈ 
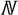
(5)

In the above model, two different charging times could be chosen at one stop, i.e., *l_i,j_* = *l_i,k_* = 1, but because Equation (1) needs to be solved for the minimum value, the case where *l_i,j_* = *l_i,k_* = 1 will not occur. As the maximum charging time at each candidate stop was *z*, only if *l_i,j_* = *l_i,k_* = 1, and *i* < *k*, Equation (2) could be satisfied. It indicates that *l_i,h_* could be selected and *k* ≦ *h* ≦ *z*, which could also satisfy Equation (2). Furthermore, if *l_i,j_* = *l_i,k_* = 1, then *l_i,j_* + *l_i,k_* + (*j* + *k*)/(*m* + 1) *z* > 1 + *h*/(*m* + 1)*z*. It can be observed from the above analysis that if *l_i,j_* = *l_i,k_* = 1, Equation (1) is not the minimum value.

According to the above model, the charging stops could be placed and the charging time at each charging stop could be set up. Moreover, the total charging time would be the shortest with the minimum number of charging stops. Equation (1) can be divided into two parts: a natural number and a decimal. As the maximum number of charging stops was *m*, and the maximum charging time at each charging stop was *z*, the minimum total charging time would not be more than *mz*. Therefore, the natural number in (1) is the number of charging points, and the decimal part multiplied by (*m* + 1)*z* is the total charging time.

## 4. Planning Strategy Based on the Lower Bound of the Optimal Solution

For model (1), the possible locations and charging time of the optimal charging stops were considered. Accordingly, Algorithm 1 was proposed to plan the charging stops. The maximum difference between the setup results and the optimal setup results was analysed. In Algorithm 1, sensor *s_i_* that consumed the most charging time *t_i_* was detected first. Subsequently, seven charging stops were placed in a circle of radius 2*r* with *s_i_* as the centre, as shown in Figure 4. The time at each charging stop was *t_i_*. Finally, sensors that were less than 2*r* from *s_i_* were removed from *S*. If there were still sensor(s) in *S*, then these steps were repeated until all the sensors in *S* were removed.

**Algorithm 1.** Planning strategy based on the lower bound of the optimal solution.Input: *S*: Sensor, *T*: Charging time of the sensorOutput: *D*: Charging stop, *C*: Charging time at the charging stop*D* = ∅While *S* ≠ ∅ doFind sensor *s_i_* ∈ *S* with the maximum *t_i_* ∈ *T*Add {*d*_1_, *d*_2_, *d*_3_, *d*_4_, *d*_5_, *d*_6_, *d*_7_} to *D*Set the charging time *c_j_* of *d_j_* ∈ {*d*_1_, *d*_2_, *d*_3_, *d*_4_, *d*_5_, *d*_6_, *d*_7_} as *t_i_*If *distance*(*d_i_* ∈ *D*, *s_k_* ∈ *S*) ≦ *r*Remove *s*’*_k_* from *S*End IfEnd While

Although the minimum total number of stops and the minimum total charging time were unknown, the planning results of Algorithm 1 could be obtained. The number of sensors and the total charging time would not be more than seven times the minimum number of stops and the minimum total charging time, respectively. If a charging stop needed to be set up to charge sensor *s_i_*, it had to be within the circle of radius *r* with sensor *s_i_* as the centre. Moreover, sensor *s_j_* that could be charged simultaneously with *s_i_* would be in the circle of radius 2*r* with sensor *s_i_* as the centre. According to the setup shown in Figure 4, seven charging stop locations could be set up, such that the sensor *s_j_*, which could be charged simultaneously with *s_i_*, would be in the charging range of at least one charging stop. It is known from the above analysis that in each iteration, a sensor that was not within the charging range of the charging stops would be selected by Algorithm 1. Moreover, the distance between the two sensors selected from two iterations would be more than 2*r*. The minimum number of charging stops would be at least the number of iterations of Algorithm 1. In each iteration, Algorithm 1 would set up seven charging stops. Thus, the number of charging stops would not exceed seven times the minimum number of charging stops. The charging time *t_i_* required by sensor *s_i_* selected by each iteration would be larger than the charging time *t_j_* required by sensor *s_j_*, which could be charged simultaneously with *s_i_*. The sum of charging times at all the charging points in the circle of radius 2*r* with *s_i_* as the centre could not be less than *t_i_*. The distance between two sensors selected by any two iterations would be more than 2*r*. The charging time of sensor *s_i_* selected from each iteration was *t_i_*. The sum of *t_i_*, which was the charging time of sensor *s_i_* selected by each iteration, would be the lower bound of the time required for charging all the sensors. In each iteration, the sum of charging time set by Algorithm 1 was 7*t_i_*. Therefore, the results of Algorithm 1 could be considered as seven times the lower bound of the time required for charging all the sensors.

## 5. An Improved Strategy for Planning Charging Stops by Minimum Clique Partition

When planning charging stops, the sensors could be divided into cliques. The goal was to have the least number of cliques and reduce the number of stops required by each clique of sensors. If the number of sensor cliques is larger, the number of stops required by each clique of sensors would be less. However, a clique of sensors would need at least one stop. It there are too many cliques of sensors, even if only one stop was needed by each clique, the total number of stops would be too high. On the contrary, if the number of sensor cliques was too small, one clique of sensors may need multiple stops. Consequently, many stops may be generated. An ideal grouping result is that each clique of sensors only needs one charging stop while having the minimum number of sensor cliques. 

The use of an auxiliary graph to assist in grouping the nodes in a sensor network is often adopted. However, a simple carelessness in designing an auxiliary graph may result in sub-optimal solutions. In the case of deploying wireless chargers, this means an excessive number of selected charging stops. In general, an auxiliary graph *G*(*V*,*E*) was established and the minimal clique algorithm on the auxiliary graph was used as the sensor grouping strategy. If the unit disk graph *G*(*V*,*E*) was established according to the relationship between the sensor location and the charging distance, the location of sensor *s_i_* ∈ *S* was the point *v_i_* ∈ *V*, and the edge *e_i_*_,*j*_ ∈ *E* indicated that sensors *s_i_*, *s_j_* could simultaneously receive power from one stop, according to the setup method described by Khelladi et al. [2] and Lin [8]. As the charging distance was *r*, *e_i_*_,*j*_ also indicated that the distance between sensors *s_i_* and *s_j_* would not exceed 2*r*. Therefore, the aforementioned *G*(*V*,*E*) is a unit disk graph. After the unit disk graph was established to represent the relationship between sensors, the charging stops could be studied and planned using the minimal clique on the unit disk graph. Through the related research on the unit disk graph, the sensors could be partitioned into cliques, and the minimum clique numbers could be approximated, such that the distance between any two sensors in each clique would not exceed 2*r*. This also indicated that any two sensors in the same clique could receive power from the same charging stop. 

When the distance between any two sensors in a clique did not exceed 2*r*, it may seem unlikely that multiple charging stops would be required. However, as the number of sensors in a clique increased, the probability of requiring multiple stops for a clique could also be higher. This plan first explored the maximum number of charging stops required by a clique, and the probability that a clique would require at least two charging stops. Accordingly, the feasibility of using the auxiliary graph as the strategy for planning charging stops was considered. 

Although the distance between any two sensors in a clique would not exceed 2*r*, a clique would require at most three circles of radius *r* in order to be covered. This indicated that (at most) three charging stops need to be set up for a clique. Considering Figure 5 as an example, the three sensors *s_i_*, *s_j_* and *s_k_* were distributed as an equilateral triangle with a side length of 2*r*. An intersection region could be obtained by taking the locations of sensors *s_i_*, *s_j_* and *s_k_* as the circle centre and drawing circles with a radius of 2*r*. In this intersection, the distance between any two sensors would not exceed 2*r*, and the sensors in this intersection region could be considered as one clique. To cover the arc *s_i_s_j_* and arc *s_i_s_k_* areas with circles of radius *r*, two circles were required for complete coverage. The intersection points of the two circles that covered arc *s_i_s_j_* and arc *s_i_s_k_* would be within arc *s_j_s_k_*. Therefore, two circles with a radius of *r* would not fully cover the arc *s_j_s_k_* area. As the charging range of the wireless charger was a circle with a radius of *r*, it is known from the above analysis that, even if two charging stops were set up for sensors in the same clique, they may not necessarily be able to charge every sensor in the clique. 

Although setting up two charging stops may still not be sufficient to supply power to all the sensors in the same clique, it can be guaranteed that three charging stops would be able to provide power to every sensor in the same clique. First, it is known that the furthest distance between any two sensors *s_i_* and *s_j_* in a clique would not exceed 2*r*. Therefore, if an intersection region was established by using *s_i_* and *s_j_* as the circle centre and 2*r* as the radius, the clique could be confined within this intersection region. As shown in Figure 6, by drawing a boundary *B* parallel to and of distance *r* below the horizontal line segment *s_i_s_j_* in the intersection region, the grouping could subsequently be confined in region *R* above *B* in the intersection region. If there was an *s_a_* outside *R*, and the distance between *s_a_* and the sensor *s_b_*, which was furthest from it, was 2*r*, *s_b_* would be below the boundary *B*’ drawn parallel to and of distance *r* above the line segment *s_i_s_j_*. The clique could still be covered by a region similar to *R*.

As shown in Figure 7, it can be observed that three circles with a radius of *r* could completely cover the region *R*. Therefore, after completing grouping, each clique would need the setting up of at most three charging stops to charge every sensor. 

To understand the case where a clique needs the setting up of more than one charging stop, a simulation experiment was employed to generate different number of sensors in a clique and examine whether the sensors in a clique could be covered by a circle of radius *r*. The probability that a clique requires more than one charging stop was determined. The method of clique generation was introduced first, and subsequently, the statistical results were shown. The cliques were generated by randomly deploying sensors sequentially. The clique size *u* and the clique condition *v* were the parameters. The parameter *u* represented the number of sensors in a clique *Q*. The parameter *v* indicated that the distance between any two sensors in the clique did not exceed *v*. First, a sensor *p* was generated randomly, and the distance between *p* and *q* ∈ *Q* was checked. If the distance between *p* and *q* exceeded *v*, another sensor *p*’ was generated. If the distance between *p* and every sensor in *Q* was less than *v*, *p* was added to *Q* until |*Q*| = *u*.

According to the above deployment method, different parameter combinations were tested. Each group of parameters generated 1000 cliques. The probability of requiring more than one charging stop was analysed statistically, as shown in Figure 8. The parameter *u* was set from 1 to 100, and *v* was 1, 5, 10, 15, 20, and 25. It can be observed from Figure 7 that the probability of requiring to set up more than one charging stop increased as the number of sensors in the clique increased. It was not affected by the clique conditions. When the number of sensors in a clique was 10, the probability of requiring to set up more than one charging stop was approximately 50%. When the clique size increased to 20, the probability of requiring to set up more than one charging stop reached 80%. It is known from the above statistical results that the larger the clique size, the larger is the probability of requiring to set up more than one charging stop. 

The number of sensors in a clique was not only affected by the distribution density of the sensors, but also by the clique conditions. When the sensors were sparsely distributed, if the charging range was large, it indicated that the clique condition was large. There would be many sensors in a clique. When the charging range was small, but the sensors were densely distributed, there would also be many sensors in a clique. It is known that both increasing the charging range and increasing the sensor distribution density would result in an increase in the number of charging stops, which would be increased to three times the number of cliques at most. 

To avoid the situation where a clique requires multiple stops, a feasible way is to change the way the unit disk graph is established. As mentioned earlier, the distribution of the three sensors *s_i_*, *s_j_*, and *s_k_* was an equilateral triangle with a side length of 2*r*. It could not be covered by a circle with a radius of *r*, but could be covered by a circle with a radius of $2r\cdot \sqrt{3}/2$. This indicated that, if the farthest distance between sensors in the clique did not exceed *q*, each clique only required the setting up of one charging stop. However, if the value of *q* became smaller, the number of cliques may also increase. Therefore, the maximum value of *q* should be determined, such that the distance between any two sensors in a clique does not exceed *q*, and each clique only needs the setting up of one charging stop. 

If each clique could only be covered by a circle with a radius of *r*, the distance between any two sensors in a clique could not exceed $\sqrt{3}$*r*. The distance limit between sensors could be speculated through the characteristics of the minimal enclosing circle and the sine theorem. The radius of the minimal enclosing circle could be obtained by at most three points, *s_i_*, *s_j_* and *s_k_*. If the radius is obtained from *s_i_* and *s_j_*, the radius of the minimal enclosing circle is the distance between *s_i_* and *s_j_*. If the radius is obtained from *s_i_*, *s_j_* and *s_k_*, the minimal enclosing circle is the circumcircle of the acute triangle formed by *s_i_*, *s_j_* and *s_k_*. As *s_i_*, *s_j_* and *s_k_* form an acute triangle, if ∠*s_i_s_j_s_k_* is the largest angle of the triangle, then 60° ≤ ∠*s_i_s_j_s_k_* < 90°. According to the aforementioned characteristics of the minimal enclosing circle and the sine theorem, if the distance between *s_i_* and *s_k_* is $\sqrt{3}$*r*, and ∠*s_i_s_j_s_k_* = 90°, the radius of the minimal enclosing circle of *S* is *r*. Therefore, when grouping the sensors by the minimal clique of the unit disk graph, if the edge *e_i_*_, *j*_ indicates that the distance between *s_i_* and *s_j_* does not exceed $\sqrt{3}$*r*, then each clique only needs the setting up of one charging stop.

## 6. 2-Phase Setup Strategy

In this section, a 2-phase setup strategy was proposed for the charging stops planning problem. First, the charging stops were set up with the objective of minimising the number of charging stops. Subsequently, the charging time at each charging stop was allocated, so that the total charging time was minimised and every sensor could receive sufficient power. Before proposing the setup strategy, the charging stop planning problem was first divided into two sub-problems, and subsequently, the planning method was proposed in order. 

In the proposed charging stops planning model, if *t_j_* in Equation (2) is changed to 1, then (1) only indicates setting the minimum number of charging stops, and the charging stops planning model could be simplified. The planning model with the objective of minimising the number of charging stops can be expressed as follows: (6)$$\min\overset{m}{\underset{i=1}{\sum }}l_{i}$$
subject to:(7)$$\overset{m}{\underset{i=1}{\sum }}a_{ij}l_{i}\geq 1$$
∀*j* ∈ {1, …, *n*}(8)
*l_i_* ∈ {0, 1}(9)

After setting up the charging stops, the charging time at every charging stop was set, and the objective was to have the minimum total charging time. After setting the charging stops *D* = {*d*_1_, …, *d_k_*}, the charging time at every charging stop *d_i_* was set to *c’_i_*. The sum of *c’_i_* was minimised, such that the sum of *c’_i_* at *d_i_*, which was less than distance *r* from *s_j_* was not less than *t_i_*. The model for minimising the total charging time is described as follows:(10)$$\min\overset{k}{\underset{i=1}{\sum }}c\prime_{i}$$
subject to:(11)$$\overset{k}{\underset{i=1}{\sum }}a_{ij}c\prime_{i}\geq t_{j}$$
∀*j* ∈ {1, …, *n*}(12)
*c’_j_*, *t_i_* ∈ 
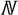
(13)

First, the method of setting up the charging stops locations was introduced. As the power receiving range of every sensor was the same, the charging ranges of the charging stops could be regarded as the same unit circle. The sensors were the targets to be covered by the unit circle. If the objective was only to have the minimum number of charging stops, the algorithm of minimal unit disk cover could be used. In the first phase of the 2-phase setup strategy, an 5-approximation algorithm [23] was adopted to plan the locations of the charging stops. 

According to the relationship between the minimal unit disk cover problem and the minimum number of charging stops problem, the charging stops were set up as follows. First, the sensor distribution area was divided into hexagons of side length *r*, as shown in Figure 9. Subsequently, hexagons with sensors were selected in the area, and the centre points of the hexagons were set as the charging stops. According to the aforementioned stops setup method, it could be ensured that the number of charging stops would not exceed five times the minimum number of charging stops. 

According to the setup results in the first phase, Algorithm 2 was proposed to set the charging time at the charging stop, and subsequently, the maximum difference between the results of Algorithm 2 and the minimum charging time was discussed. Let the charging stops set up previously be *D* = {*d*_1_, …, *d_k_*} and let *distance*(*d_i_*, *s_j_*) represent the distance between charging stop *d_i_* and sensor *s_j_*. First, sensor *s_j_*, which required the most charging time *t_j_*, was determined. Subsequently, if the distance between *d_i_* and *s_j_* does not exceed *r*, then *c_i_* = *t_j_*. After setting the charging time at the charging stops, if the distance between sensor *s_k_* and the stop at which the charging time has been set does not exceed *r*, then *s_k_* is removed from *S*. After performing the above removal steps, if |*S*| > 0, then return to the first step until |*S*| = 0. After allocating the charging time at every charging stop using the above algorithm, every sensor could satisfy the charging time requirement. 

**Algorithm 2.** Charging time setup strategy.Input: *D*: Charging stop, *S*: Sensor, *T*: Sensor charging timeOutput: *C*: Charging time at the charging stopSort *S* by *T* from large to small to form *S*’While *S*’ ≠ ∅ doIf *distance*(*d_i_* ∈ *D*, *s*’*_j_* with the largest *t_j_*) ≦ *r* and *c_i_* = 0Let *c_i_* ∈ *C* be *t_j_*;End IfIf *distance*(*d_i_* ∈ *D*, *s’_j_* ∈ *S*’) ≦ *r* and *c_i_* ≠ 0Remove *s*’*_j_* from *S*’End IfEnd While

After introducing the charging time setup strategy, the worst setup result of the setup strategy is analysed below. First, according to the charging stop setup strategy, it is known that a sensor could receive power from at most three charging stops. Considering Figure 10 as an example, let the hexagon with *d_a_* as the centre be adjacent to the hexagons with *d_b_*, *d_c_*, *d_e_*, *d_f_*, *d_g_* and *d_h_* as the centres. If there is a point *p* in the circular area centred at *d_a_*, *d_b_*, *d_c_* and *d_e_* simultaneously, because the distance between *d_e_* and *d_b_* is 3*r*, the radius of the circular area centred at *d_e_* or *d_b_* must be at least 1.5 *r*. The circumcircle of hexagon centred at *d_a_* intersected with the circumcircles of hexagons centred at *d_b_*, *d_c_*, and d*_e_*. The circumcircle of hexagon centred at *d_c_* intersected with the circumcircles of hexagons centred at *d_b_* and *d_e_*. Therefore, it can be observed that there is an intersection point between the circumcircle of hexagon centred at *d_a_* and the circumcircles of hexagons centred at *d_b_* and *d_c_*.

By combining the characteristic that a sensor could receive power from at most three charging stops and the setup method of Algorithm 2, the worst setup result of Algorithm 2 could be obtained. If sensor *s_i_* requires the longest charging time *t_i_*, according to the charging stop setup strategy, it is known that the distance between s*_i_* and the three charging stops *d_a_*, *d_b_* and *d_c_* would not exceed *r*, as shown in Figure 10. Therefore, the sum of the charging times at *d_a_*, *d_b_* and *d_c_* would not be less than *t_i_*. However, Algorithm 2 would let the charging time at *d_a_*, *d_b_*, and *d_c_* be *t_i_*. In addition, if sensor *s_i_* with the longest charging time *t_i_* selected from each iteration of the algorithm was added to the set *U*, then neither of the two sensors in *U* could be charged simultaneously by the charging stop *d_i_*. Therefore, it is known that a charging time larger than 0 must be set for |*U*| charging stops. From the above analysis, it is known that the worst result from Algorithm 2 was three times the minimum total charging time. 

The time complexity of the 2-phase setup strategy proposed in this paper is O(*n*log*n*). In the first phase, the sensor *s_i_* in a hexagon could be known each time the hexagon was selected. The time complexity for selecting all the charging stops was linear time [23]. Before setting the charging time at charging stops, if the charging time required by sensors was sorted first, the charging time could be set for each charging stop according to the sorting result. When setting up the stops, the sensors within its charging range were known, and hence, after setting the charging time, the sensors in the charging range could be directly removed and there was no need to search again. Thus, in the phase of setting the charging time at the charging stop, the time consumed is O(*n*log*n*). The time complexity for the 2-phase steps was O(*n*) and O(*n*log*n*), respectively. 

## 7. Simulation Experiment

Before introducing the simulation experiment results, we will explain the comparison method used in this study. In planning the deployment of wireless chargers, most studies adopted various types of constraints to achieve different charging objectives. In this paper, we compared our method with MSP [2] and DPH [10]. The charging objective, the sensor model, and the charging model of MSP are the most similar the ones used in this paper. DPH considered minimizing the number of charging stops, however, the charging stops were selected for an optimal 3D flight path of drones. Therefore, in the comparison we limited the maximum height used by DPH. Besides comparing the MSP method and DPH proposed in the past studies [2,10], the construction of unit disk graph was changed, ensuring each clique did not need the setting up of additional charging stops. It was expressed as Clique(1.732*r*) in the simulation experiment results. In addition, the 2-phase setup strategy, linear programming, and greedy algorithm were also compared. Except for the greedy algorithm, the other methods have been detailed above. The greedy algorithm and the experimental parameter setting will be introduced below. 

### 7.1. Simulation Experiment Setup

In the greedy algorithm, we progressively selected the location where the most sensors can be charged as the charging stop. To simplify the selection of charging stops, the sensor deployment area was first divided into a grid. Each grid point *o_i_* ∈ *O* was a candidate location of the charging stop. The number of sensors in the charging range of *o_i_* was subsequently checked. The *o_i_* with the most sensors within its charging range was identified and selected as the charging stop. Subsequently, sensor *s_k_* within the charging range of *o_i_* was removed from *S*. If |*S*| > 0, the above selection method was continued until |*S*| = 0.

After introducing the simulation experiment comparison method, the experimental parameter settings are explained subsequently. Referring to the parameter setting of [2], the sensors were deployed in a randomly distributed manner in a square area with a side length of 25 m. The power receiving range of the sensor was 2.7 m. In the experiment, the number of deployed sensors was set as the variable, with an interval of 100. A total of 10 groups of variables from 100 to 1000 were set. Based on the above variables, the number of charging stops set up using different methods was compared. In addition to comparing the number of charging stops, the charging time at the charging stop was compared. Thus, the charging time required by each sensor was determined randomly. The upper limit of the charging time required for the sensor was set as the variable, with an interval of five units of time. A total of 5 variables from 5 to 25 were set. For the aforementioned parameters, 100 experiments were performed for each group. The results were statistically averaged. 

### 7.2. Number of Charging Stops

According to the above parameter settings, the simulation experiment results of the number of charging stops are shown in Figure 11. Setups with different numbers of sensors were compared. When the number of sensors was above 200, the proposed 2-phase setup strategy was better than MSP. As the number of sensors increased, the proposed method achieved results closer to those of linear programming, and the setting number did not increase. Compared with the clique-based setup method, the upper limit of the setting number of the proposed 2-phase setup strategy did not depend entirely on the number of sensors. It was also affected by the size of the sensor deployment area. Therefore, if the sensors were randomly distributed, when the sensor distribution density was sufficiently large, every hexagon would be regarded as a charging stop; hence, the number of charging stops would not increase. However, MSP did not consider the size of the sensor deployment area, which led to the conclusion that the upper limit of the number of charging stops was only affected by the number of sensors. This resulted in an increase in the number of charging stops with the increase in the number of sensors. On the other hand, although DPH adjusted the locations of charging stops by the distribution of sensors, in determining the required number of charging stops, it still preferred to use more stops than necessary whenever possible.

The results shown in Figure 11 also corresponded with the results shown in Figure 7. Although reducing the clique condition from 2*r* to $\sqrt{3}$*r* would increase the number of cliques, the situation where several charging stops had to be set up for each clique could be avoided. However, when the number of sensors increased, if 2*r* was used as the clique condition, the clique size was also likely to increase. Therefore, most cliques required the setting up of more than one charging stop, which resulted in a rapid increase in the number of charging stops. In Figure 11, when the number of sensors exceeded 300, if 2*r* was used as the clique partition condition, the number of charging stops would exceed that of the setting method where $\sqrt{3}$*r* is used as the clique partitioning condition. 

### 7.3. Charging Time at Charging Stops

When the upper limit of the charging time required by a sensor was 25 h, the charging time at the charging stop is shown in Figure 12. The trend shown in Figure 12 is similar to that shown in Figure 9. Except for the proposed method, in both the greedy algorithm and the clique-based setup, the charging time increased significantly with an increase in the number of sensors. However, more charging stops would not necessarily increase the charging time at the charging stops. If there were intersections between the charging stop sets, the total charging time could be reduced by allocating charging time at different charging stops. However, MSP did not consider how to allocate the charging time at the charging stops effectively, which caused the charging time to increase with the number of charging stops. Although the charging time at charging stops was set directly in order to analyse the worst-case scenario, when the number of sensors was above 200, owing to the small number of charging stops, there would also be less charging time. 

In Figure 13, different upper limits for the charging time of sensors were set and the differences between the MSP and proposed methods were compared. To show the differences between the different methods, the results obtained from the proposed method were divided by the results obtained from Khelladi et al. [2]. 

First, it can be observed that the ratio between different methods was not affected by the upper limit of charging time. Subsequently, when there were 100 sensors, the charging time of the proposed method approached 1.2 times that of the MSP method. However, when the number of sensors increased to 200, the charging time of the proposed method was only 80% of that of the MSP method. Furthermore, when the number of sensors increased to 1000, the charging time of the proposed method was reduced to 40% of that of the MSP method. 

Figure 14 shows the relationship between the number of charging stops and the charging time. It can be observed that the proposed method and the MSP method exhibited the same differences in charging time and the number of stops. The difference was obtained by dividing the number of stops generated in the proposed method by that obtained from the MSP method. It can be observed from Figure 13 that the difference between the numbers of stops was consistent with the difference between charging times. For both the charging time and the number of charging stops, when the number of sensors was 200, the objective could be achieved with 80% of that proposed in Ref. [2]. When the number of sensors increased to 1000, only 40% of that of the MSP method was required.

## 8. Conclusions

This paper discussed the wireless charging planning problem and considered both the number of charging stops and the total charging time. First, we proposed the minimisation of the charging time based on the unit disk model. Subsequently, the minimisation of charging time was divided into the problem of minimum number of charging stops and minimum charging time allocation. For the minimum number of charging stops problem, the charging stops setup based on cliques was analysed first. It is known through analysis that, if the charging distance was *r* and the distance between any two sensors in a clique did not exceed 2*r*, the above clique condition would cause a clique to require at most three charging stops. Moreover, the larger the clique size, the more likely it was for a clique to require more than one charging stop. To understand the relationship between the probability of the above case to occur and the clique size, experiments were conducted to analyse the probability statistically. From the results, it was inferred that the clique-based charging stop setup method was suitable for scenarios where the sensor deployment density was sparse and the charging range was small. Subsequently, the sensor deployment area was divided into several non-overlapping hexagons, which were used as the candidate stops to plan the setup locations of the charging stops. When setting the charging time at charging stops, as the deployment area was divided into non-overlapping hexagons, we can analyse the maximum difference between the total charging time of the setup and the minimum charging time based on this feature. 

For the minimum number of charging stops, the worst-case scenario of the 2-phase setup method was analysed. In addition, it was observed through simulation experiments that, when the sensor deployment number increased, the results of the 2-phase setup method would gradually approach the minimum number of charging stops. However, the charging stops from the clique-based setup method (MSP) would increase with the number of deployed sensors, and the clique size would also increase, which resulted in an increase in cases where a clique would require more than one charging stop. 

After planning the charging stops, as different sensors required different charging times, if the charging time at every charging stop was appropriately allocated, the total charging time could be reduced. Although MSP set up the charging time at every charging stop, it did not allocate the charging time at charging stops from a global point of view, and hence, when the number of deployed sensors increased, both the number of charging stops and the total charging time increased. The number of charging stops and the charging time obtained from the proposed method also increased with the increase in the number of sensors, but the increase was significantly slower compared with that in MSP. It was observed through experiments that when the sensors were densely deployed or the charging range was wide, the proposed method was better than MSP both in terms of the number of charging stops and the charging time. Furthermore, the time complexity of the proposed algorithm O(*n*log*n*) was better than that O(*n*^3^) of MSP. 

## Figures and Tables

**Figure 1 sensors-19-00201-f001:**
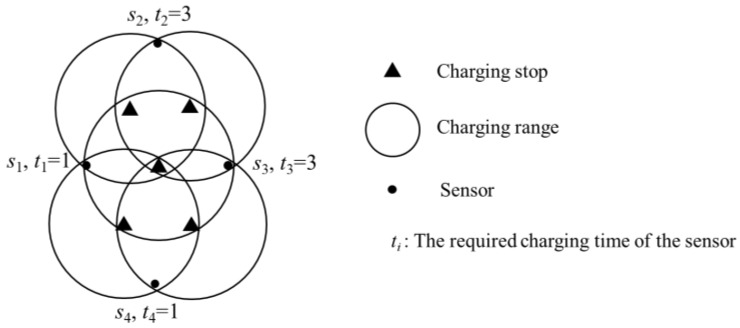
Case where the same number of stops are set up, but the total required charging time is different.

**Figure 2 sensors-19-00201-f002:**
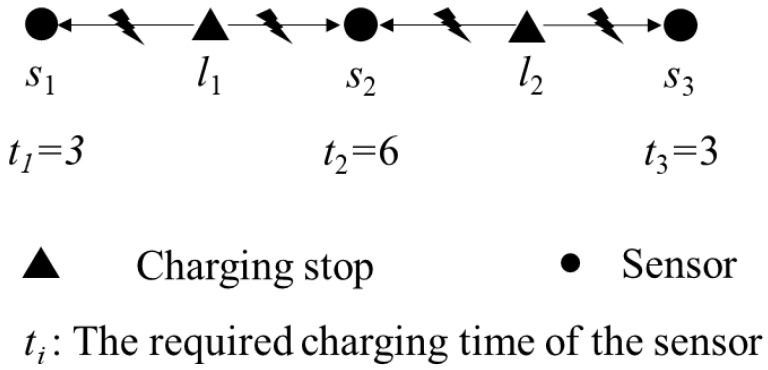
Example of charging time allocation.

**Figure 3 sensors-19-00201-f003:**
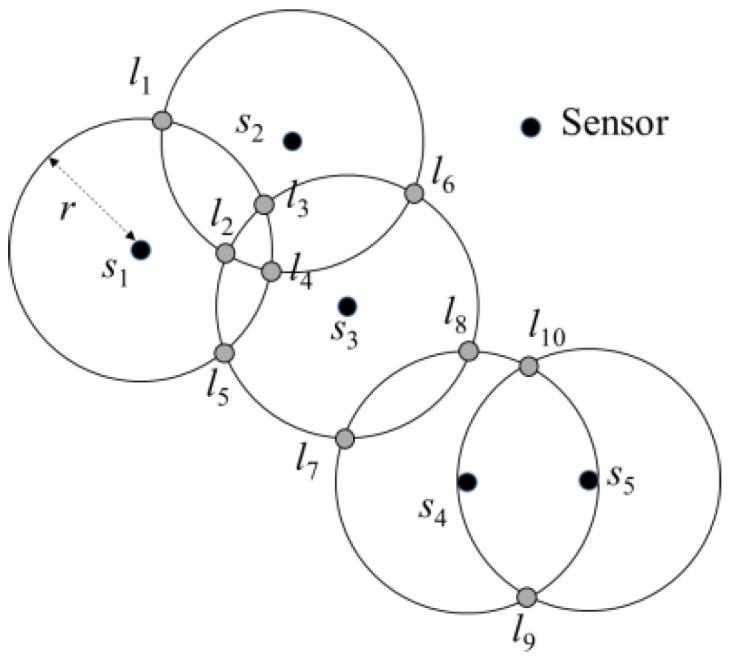
Example of candidate stops setup.

**Figure 4 sensors-19-00201-f004:**
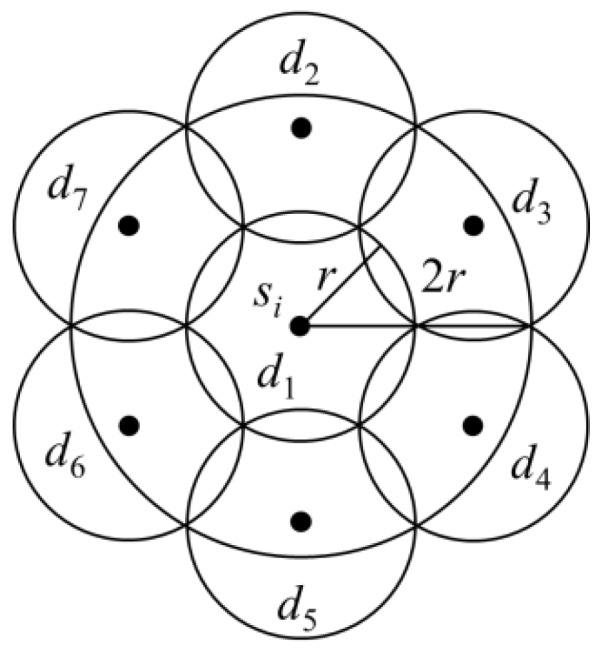
Set up locations of charging stops.

**Figure 5 sensors-19-00201-f005:**
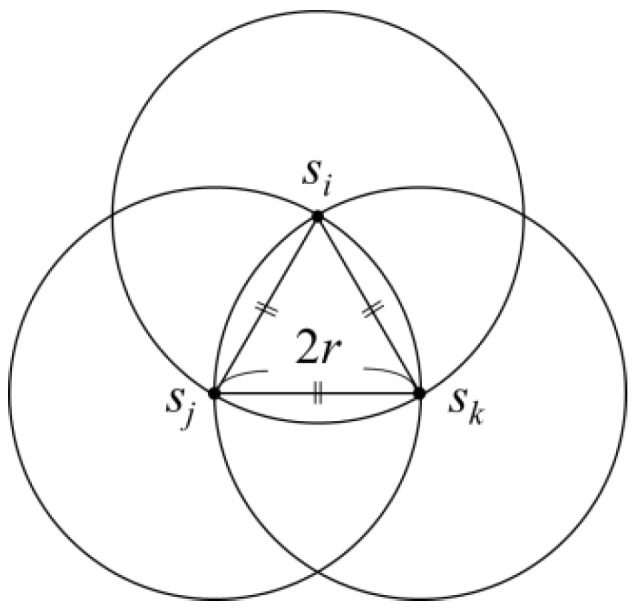
Intersection region of circles drawn with the locations of sensors *s_i_*, *s_j_*, and *s_k_* as the centres with a radius of 2*r* could not be covered by two circles with a radius of *r*.

**Figure 6 sensors-19-00201-f006:**
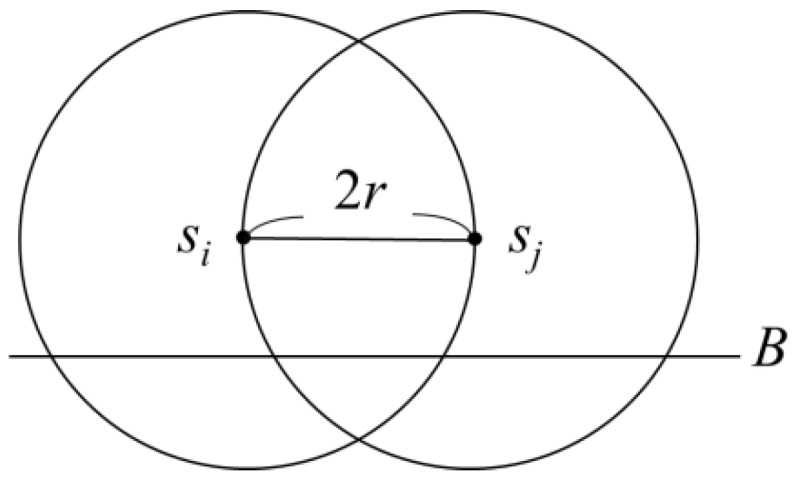
Confined region covering a clique.

**Figure 7 sensors-19-00201-f007:**
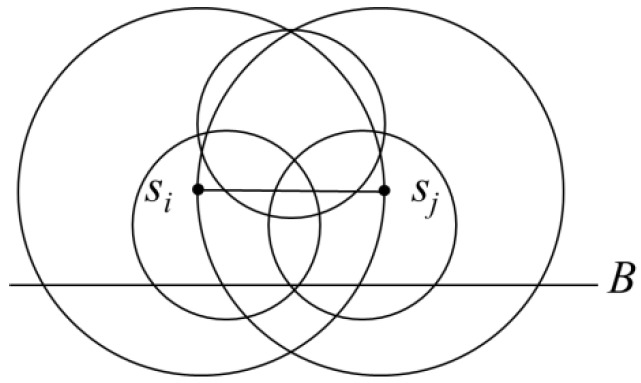
Clique covered by three circles with a radius of *r* and with a distance within 2*r.*

**Figure 8 sensors-19-00201-f008:**
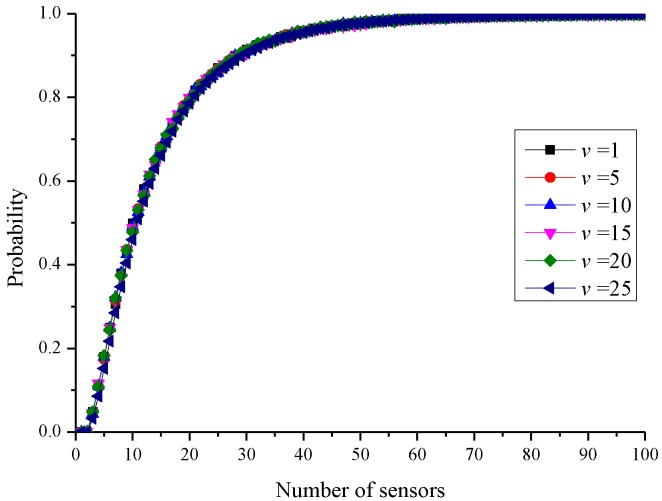
Probability of requiring to set up more than one charging stop for a clique.

**Figure 9 sensors-19-00201-f009:**
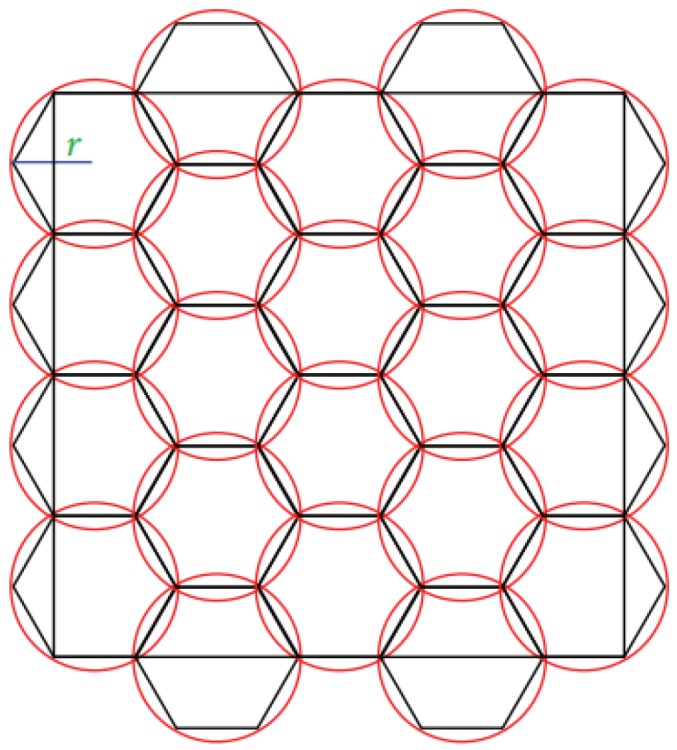
Candidate charging stops setup.

**Figure 10 sensors-19-00201-f010:**
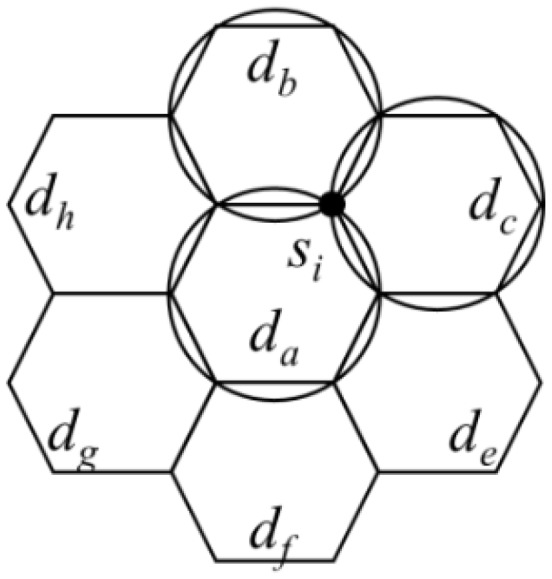
Illustration showing that a sensor could receive power from at most three charging stops.

**Figure 11 sensors-19-00201-f011:**
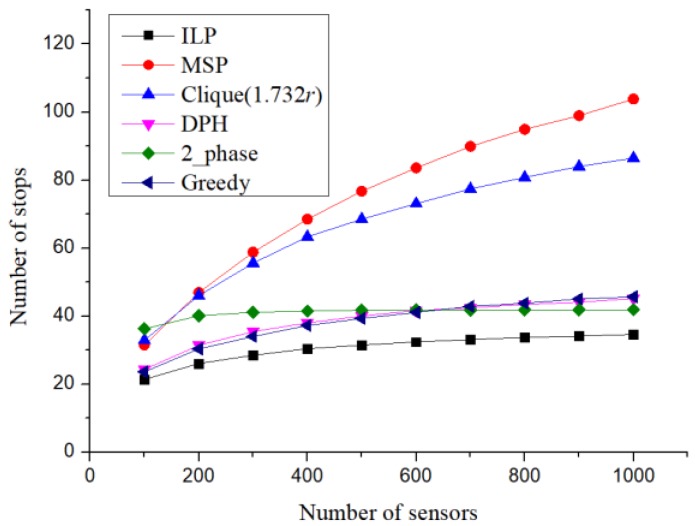
Number of charging stops.

**Figure 12 sensors-19-00201-f012:**
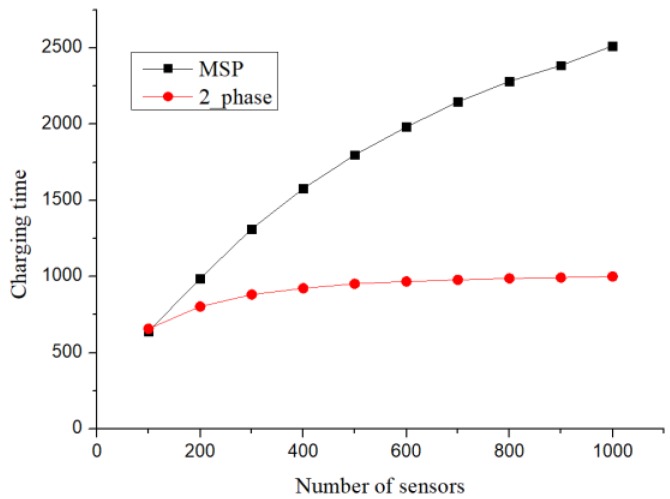
Charging time of charger.

**Figure 13 sensors-19-00201-f013:**
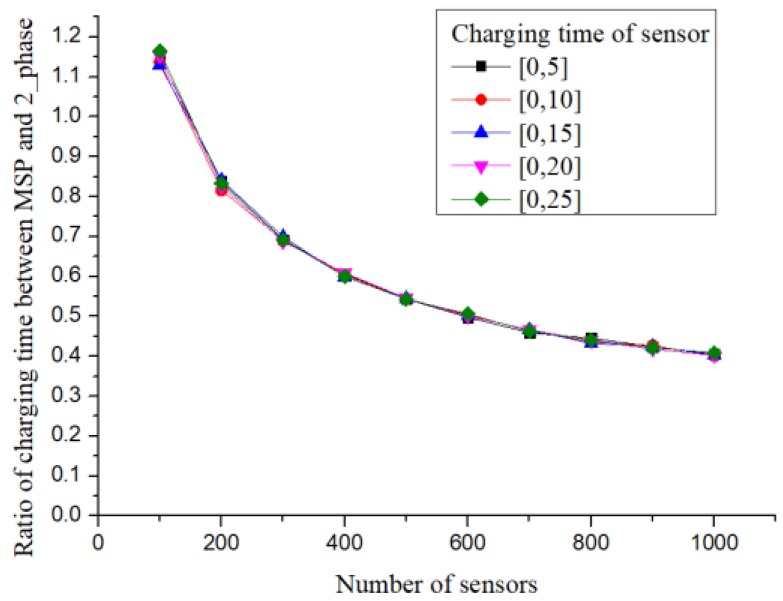
Difference ratio of charging time.

**Figure 14 sensors-19-00201-f014:**
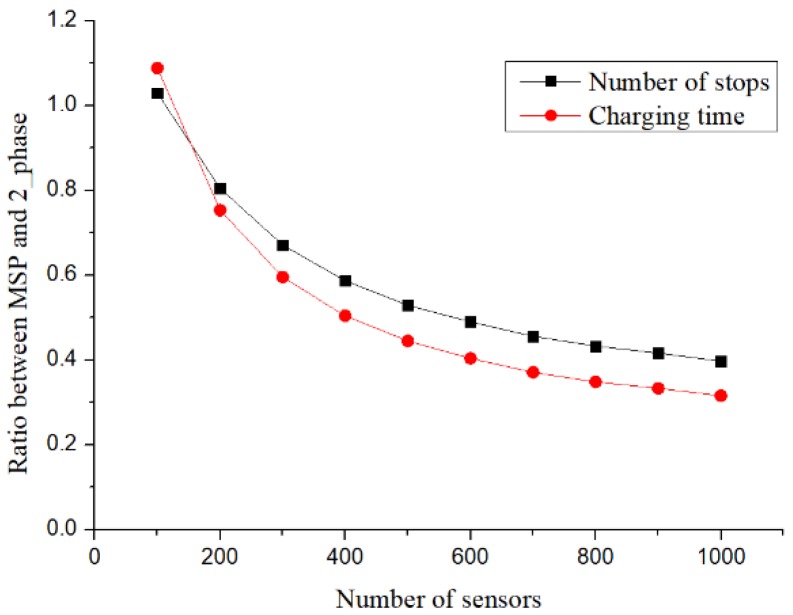
Charging time and number of stops.

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
