# Peer review of "Wireless Charging Deployment in Sensor Networks"

_sensors, 2019, doi:10.3390/s19010201_

Reviewer 1 Report

This is a good paper in a good area. It is interesting to read it and the authors have taken the effort to help the reader follow the paper.

I have two basic comments: the first is that I didn't like the abstract. It looks like written the last minute, uses a past tense that is not appropriate (to my humble opinion) for an abstract.

The second comment is that the literature review is inadequate. I understand that a few people have tackled the same problem but there are numerous people working in the area and their works even though not 100% related to this one have contributed to the area. It is most inappropriate that references are just 10. 

Author Response

Point 1: I have two basic comments: the first is that I didn't like the abstract. It looks like written the last minute, uses a past tense that is not appropriate (to my humble opinion) for an abstract.

Response 1: Thank you for your comment, we have rewritten the abstract.

Point 2: The second comment is that the literature review is inadequate. I understand that a few people have tackled the same problem but there are numerous people working in the area and their works even though not 100% related to this one have contributed to the area. It is most inappropriate that references are just 10.

Response 2: In the previous manuscript only the most related work were included. As the reviewer suggested, in order to improve the completeness of the presentation, we have added more related work and revised Section 2.

Reviewer 2 Report

In this paper, the authors focus on the optimal charging deployment in wireless sensor networks. The numbers of the stops and the charging time are two major factors to be considered. The simulation shows that the proposed methods could charge sensors with 40% of stops and charging time. 

The following issues should be addressed:

The introduction on background knowledge is not adequate. More related contents should be added. Especially, the definition on charging stops and charging time should be given formally.

In Figure 1, the authors didn't give clear instruction on the charging process. The brief introduction on the circles and the four rectangles should be be given.

In section 5, the auxiliary graphs is used for achieving minimum cliques. The authors should explain the advantage and reason for using this algorithm. Moreover, the formal definition should be presented as preliminaries in section 3.

The related work need to be improved, more relevant state-of-the-art research should be introduced. For example:

DOI: 10.1109/TNET.2014.2303979  DOI: 10.1155/2017/6596943   DOI: 10.3390/s18113930

The comparison should be improved. The proposed scheme should be compared with the existing methods. Furthermore, the line representing number of stops in Figure 14 is covered. It should be presented clearly.

Lots of typos exits, the authors should check the paper carefully. 

Author Response

Point 1: The introduction on background knowledge is not adequate. More related contents should be added. Especially, the definition on charging stops and charging time should be given formally.

Response 1: Thank you for the suggestion, we have supplemented the explanation of charging stops and charging time in Section 1.

Point 2: In Figure 1, the authors didn't give clear instruction on the charging process. The brief introduction on the circles and the four rectangles should be given.

Response 2: We have redrawn Figure 1 and Figure 2 and fixed the labels to improve their clarity.

Point 3: In section 5, the auxiliary graphs is used for achieving minimum cliques. The authors should explain the advantage and reason for using this algorithm. Moreover, the formal definition should be presented as preliminaries in section 3.

Response 3: We have added explanations and changed the title of Section 5 to better address the purpose of the use of an auxiliary graph.

Point 4: The related work need to be improved, more relevant state-of-the-art research should be introduced. For example:

DOI: 10.1109/TNET.2014.2303979 

DOI: 10.1155/2017/6596943 

DOI: 10.3390/s18113930

Response 4: We have revised Section 2 and expanded the references according to your valuable suggestion.

Point 5: The comparison should be improved. The proposed scheme should be compared with the existing methods. Furthermore, the line representing number of stops in Figure 14 is covered. It should be presented clearly.

Response 5: We have revised Section 7. The optimization objective is different in every study. The optimization objective and the charging model used by MSP [1] is most comparable to our work. [2] considered minimizing the number of charging stops and proposed the DPH method. We have adjusted their corresponding parameters for the fairness of comparisons.

1. Khelladi,L.; Djenouri, D.; Rossi, M.; Badache, N. Efficient on-demand multi-node charging techniques for wireless sensor networks.Comput. Commun. 2017, 101, pp. 44–56.

2. Zorbas, D; Douligeris, C. Computing optimal drone positions to wirelessly recharge IoT devices. IEEE Conference on Computer Communications Workshops. 2018, pp. 628-633.

Point 6: Lots of typos exits, the authors should check the paper carefully.

Response 6:Thank you for your kind reminder. We have carefully checked the paper to remove the typos.

Round  2

Reviewer 2 Report

After revision, this paper is good enough to be published in Sensors journal.